# Efficacy of a Protein Vaccine and a Conjugate Vaccine Against Co-Colonization with Vaccine-Type and Non-Vaccine Type Pneumococci in Mice

**DOI:** 10.3390/pathogens9040278

**Published:** 2020-04-10

**Authors:** Gabriela B. C. Colichio, Giuliana S. Oliveira, Tasson C. Rodrigues, Maria Leonor S. Oliveira, Eliane N. Miyaji

**Affiliations:** Laboratorio de Bacteriologia, Instituto Butantan, São Paulo 05503-900, Brazil; gabriela_cherulli@hotmail.com (G.B.C.C.); giuliana.oliveira@butantan.gov.br (G.S.O.); tasson.rodrigues@butantan.gov.br (T.C.R.); marialeonor.oliveira@butantan.gov.br (M.L.S.O.)

**Keywords:** *Streptococcus pneumoniae*, co-colonization, vaccine, PCV13, PspA

## Abstract

Widespread use of pneumococcal conjugate vaccines (PCVs) has led to substitution of vaccine-type (VT) strains by non-vaccine type (NVT) strains in nasopharyngeal carriage. We compared the efficacy of PCV13 and a nasal protein formulation containing pneumococcal surface protein A (PspA) adjuvanted with the whole-cell pertussis vaccine (wP) in the protection against co-colonization challenge models in mice with VT and NVT strains expressing different PspAs. Immunized mice were challenged with two different mixtures: i. VT4 (PspA3) + NVT33 (PspA1) and ii. VT23F (PspA2) + NVT15B/C (PspA4). Results from the first mixture showed a reduction in loads of VT4 strain in the nasopharynx of mice immunized with PCV13. A statistical difference between the loads of the VT and NVT strains was observed, indicating a competitive advantage for the NVT strain in PCV13-immunized animals. In the second mixture, no reduction was observed for the VT23F strain, probably due to low levels of anti-23F polysaccharide IgG induced by PCV13. Interestingly, a combination of the PspA formulation containing wP with PCV13 led to a reduction in colonization with both strains of the two mixtures tested, similar to the groups immunized nasally with wP or PspA plus wP. These results indicate that a combination of vaccines may be a useful strategy to overcome pneumococcal serotype replacement.

## 1. Introduction

*Streptococcus pneumoniae* frequently colonizes the human nasopharynx asymptomatically, but it can also cause diseases such as sinusitis, otitis media, pneumonia, sepsis, and meningitis. High pneumococcal carriage rates have been described in children, especially in low-income countries, where colonization rates of 20% to 93.4% in children less than five years of age have been described [1]. Co-colonization with two or more pneumococcal serotypes is also common, reaching 50% in carriers [2,3].

The capsular polysaccharide (PS) is the most important virulence factor of pneumococcus. Pneumococcal conjugate vaccines (PCVs) induce antibodies against PS from serotypes included in the formulation and have considerably reduced the incidence of invasive pneumococcal disease in children. PCVs also reduce the nasopharyngeal carriage of vaccine type (VT) strains, leading to herd immunity in non-immunized individuals. The burden of pneumococcal disease remains high though, and data from 2015 have estimated 3.7 million cases of severe pneumococcal disease and 294,000 deaths caused by pneumococcal infection in children aged 1–59 months worldwide [4]. There are more than 90 different capsular serotypes described [5], and PCVs afford protection against up to 13 serotypes included in the formulation. Because of the serotype-dependent protection, the widespread use of PCVs has led to the substitution of invasive disease and colonization caused by VT strains with non-vaccine type strains (NVT), a phenomenon known as serotype replacement [6,7,8].

An alternative to overcome serotype-specific protection induced by PCVs is the use of vaccines based on protein antigens. Pneumococcal surface protein A (PspA) is an important virulence factor and is considered a promising vaccine candidate. It was shown to induce protection against colonization, pneumonia, and invasive disease models in mice [9,10,11]. PspA shows some variability and has been classified into three families: Family 1 (clades 1 and 2), Family 2 (clades 3, 4, and 5) and Family 3 (clade 6) [12]. Cross-reactivity of anti-PspA antibodies is higher within the same family [13,14]. Since most isolates express PspA from families 1 or 2, a broad coverage vaccine should include PspAs from both families [15]. 

Few papers have evaluated nasopharyngeal co-colonization using mouse models. Intranasal carriage of one strain was shown to inhibit the acquisition of a second pneumococcal strain [16]. Colonization models using up to six pneumococcal strains from different serotypes have also been tested in naïve mice. Clinical isolates and isogenic strains expressing capsules from serotypes 6B, 14, 19A, 19F, 23F, and 35B were analyzed and shown to compete during colonization, with serotype as a major determinant of competitive success [17]. Protection against colonization with a mixture of strains of serotypes 14 and 19F was also evaluated in mice immunized with a non-encapsulated inactivated whole-cell pneumococcal vaccine adjuvanted with cholera-toxin (CT). A trend towards fewer colony forming units (CFU) recovered from mice immunized with the vaccine was found, when compared to CT alone. No differences in the ratios between strains of serotypes 14 and 19F were observed between groups [17]. 

Our group has also tested the efficacy of intranasal immunization with PspA using mouse models of co-colonization with VT strains. We have shown that nasal immunization with recombinant PspA from clade 1 (PspA1) and PspA from clade 4 (PspA4) using the whole-cell pertussis vaccine (wP) as an adjuvant reduced colonization when challenge experiments were performed with a mixture of two isolates from serotype 6B expressing PspA1 and PspA4 or with a mixture of isolates of serotypes 6B (PspA clade 3 - PspA3) and 23F (PspA clade 2 - PspA2). The formulation did not lead to a pronounced increase in colonization of one isolate over the other, showing that the vaccine strategy would not favor replacement [18]. We now aim at testing the formulation against co-colonization models using VT and NVT isolates expressing different PspAs and compare nasal immunization with the protein formulation with parenteral vaccination with PCV.

## 2. Results

### 2.1. Antibody Response Induced by Immunization

Mice were immunized with three doses of saline, wP (adjuvant only) or PspA1 + PspA4 + wP intranasally and with PCV13 subcutaneously. The combination of intranasal immunization with protein and subcutaneous immunization with PCV13 was also tested ((PspA1 + PspA4 + wP) + PCV13). The induction of antibodies against PspA and PS was evaluated in serum samples. Animals immunized with PspA1 + PspA4 + wP and (PspA1 + PspA4 + wP) + PCV13 showed high IgG antibody titers against both PspA1 and PspA4 (Figure 1a,b). Mice immunized with PCV13 and (PspA1 + PspA4 + wP) + PCV13 showed induction of IgG antibodies against PS4 and PS23F, both VT PS (Figure 1c,d). Antibodies against PS15B and PS33 were not detected, as expected for NVT PS (not shown).

The presence of anti-PspA IgG and IgA antibodies was also evaluated in vaginal washes as a proxy for the presence of antibodies in the nasopharynx, since nasal immunization was shown to induce vaginal humoral response in women [19]. Both IgG and IgA anti-PspA1 antibodies were detected in vaginal washes of mice immunized with PspA1 + PspA4 + wP and (PspA1 + PspA4 + wP) + PCV13 (Figure 1e,f), indicating the induction of humoral mucosal response in mice vaccinated intranasally with the protein formulation. We did not measure anti-PS response in vaginal washes, since immunization was performed parenterally and antibody levels were already low in sera.

We next evaluated the binding of IgG serum antibodies to the surface of intact pneumococci by flow cytometry. VT and NVT isolates expressing different PspAs were tested: VT4 (PspA3), VT23F (PspA2), NVT33 (PspA1), and NVT15B/C (PspA4) (Table 1). Antibodies from animals immunized with PCV13 and (PspA1 + PspA4 + wP) + PCV13 showed binding to VT4 (PspA3) (Figure 2a), indicating that anti-PS4 IgG antibodies probably bind to the bacteria. Since no binding was observed in the group immunized with PspA1 + PspA4 + wP, anti-PspA IgG antibodies do not seem to recognize this strain. For strain NVT33 (PspA1) (Figure 2b), there was binding in the groups PspA1 + PspA4 + wP and (PspA1 + PspA4 + wP) + PCV13, indicating that recognition was through antibodies against PspA. As expected for an NVT strain, antibodies induced by PCV13 did not bind to this isolate. Surprisingly for VT23F (PspA2) (Figure 2c), no binding was detected in the PCV13 group, which might be due to low antibody titers against PS 23F (Figure 1d). There was binding in the groups PspA1 + PspA4 + wP and (PspA1 + PspA4 + wP) + PCV13, again indirectly indicating recognition by anti-PspA antibodies. Finally, for NVT15B/C (PspA4) (Figure 2d), binding was observed for PspA1 + PspA4 + wP and (PspA1 + PspA4 + wP) + PCV13. Once again, binding suggests recognition only by anti-PspA antibodies to this NVT strain.

### 2.2. Establishment of Co-Colonization Models using VT and NVT Isolates

To develop models of VT and NVT co-colonization, strains were transformed with vectors containing erythromycin (erm)- or spectinomycin (spec)-resistance cassettes flanked by sequences from *iga*, which encodes IgA1 protease. Since IgA1 protease activity is specific for human IgA1, it does not act as a virulence factor in mouse models [20,21]. Mixtures of two clones, one VT and one NVT expressing different PspAs and resistant to different antibiotics, were inoculated intranasally in mice. The bacterial load of each clone was evaluated in nasal washes after five days through plating in blood agar plates containing erm or spec. The inoculation of the mixtures of clones VT4 (PspA3) Spec 1 + NVT33 (PspA1) Erm 2 (Figure 3a) and VT23F (PspA2) Erm 2 + NVT15B/C (PspA4) Spec 2 (Figure 3b) resulted in the recovery of similar amounts of bacteria resistant to erm and spec, showing that these mixtures are suitable as co-colonization models.

### 2.3. Co-Colonization Challenge

Immunized mice were then challenged using the established co-colonization models with the mixtures of VT and NVT strains to evaluate protection against nasopharyngeal carriage. Mice were challenged with the mixture VT4 (PspA3) + NVT33 (PspA1), and bacterial loads of each strain were determined in nasal washes collected after five days (Figure 4a,b). A reduction of both strains was observed in the group immunized with the adjuvant wP only. The group immunized with PspA1 + PspA4 + wP showed a reduction of both strains compared to saline, but only NVT33 (PspA1) showed a statistically significant reduction compared to wP. Mice immunized with PCV13 showed significant decrease in the loads of VT4 (PspA3), which is expected for a VT strain. These animals also showed increases in the loads of NVT33 (PspA1) with borderline significance. As for the group immunized with the combination (PspA1 + PspA4 + wP) + PCV13, there was a significant reduction of both strains compared with PCV13 only. Immunization with wP and with PspA1 + PspA4 + wP also led to a significant reduction of both strains compared to PCV13. Paired analysis comparing VT4 (PspA3) and NVT33 (PspA1) loads in the same sample showed statistically significant difference for PCV13, but not for any of the other groups, indicating a competitive advantage for the NVT strain induced by PCV13 vaccination (Appendix A).

Immunized mice were also challenged with the mixture VT23F (PspA2) + NVT15B/C (PspA4) (Figure 4c,d). Once again, a reduction of both strains was observed in mice inoculated with the adjuvant wP only. The group immunized with PspA1 + PspA4 + wP showed a reduction in both strains compared to saline and to wP. There was no reduction in the loads of VT23F (PspA2) in mice immunized with PCV13, which is in accordance with the lack of binding of serum IgG antibodies to this strain (Figure 2c). The NVT strain seemed to have slightly higher loads in the PCV13 group compared with saline, though there was no statistical difference. Also with this mixture of strains, the group immunized with the combination (PspA1 + PspA4 + wP) + PCV13 showed a reduction of both strains compared to PCV13 only. Immunization with wP and PspA1 + PspA4 + wP also led to a significant reduction of both strains compared to PCV13. Paired analysis comparing VT23F (PspA2) and NVT15B/C (PspA4) loads in the same sample indicated statistically significant differences for wP and PspA1 + PspA4 + wP, with slightly higher levels of VT23F (PspA2) (Appendix A). These results indicate that the protection elicited by wP may affect strains differently.

## 3. Discussion

Widespread use of PCVs has led to a rapid substitution of VT strains by NVT strains in nasopharyngeal carriage. In this work, we aimed to compare the efficacy of PCV13 and a protein formulation containing PspA with the adjuvant wP in the protection against co-colonization challenge models with VT and NVT strains expressing different PspAs. These models would mimic the common situation of colonization with more than one pneumococcal serotype observed in children in high carriage burden settings. Furthermore, a model of pediatric pneumococcal carriage evaluating drivers of the emergence of new antibiotic-resistant lineages following the introduction of a vaccine targeting more common resistant types found that antibiotic pressure is the strongest driver, but carriage burden and multiple colonization can be important drivers if antibiotic pressure is lower [22]. These data indicate that models of co-colonization can be important to evaluate the emergence of vaccine-escape pneumococci resistant to antibiotics.

We constructed antibiotic-resistant strains to establish the co-colonization models, so we could distinguish VT and NVT strains in nasal wash samples from challenged mice. Since antibiotic resistance may reduce bacterial fitness in colonization [16], we tested different clones and selected the ones showing similar levels of colonization in naïve mice. This allowed us to compare the colonization of the two strains in groups receiving different immunizations. Mice were immunized and then challenged with two different mixtures of strains: i. VT4 (PspA3) + NVT33 (PspA1) and ii. VT23F (PspA2) + NVT15B/C (PspA4). Results from the first mixture showed a reduction of the serotype 4 VT strain and a slight increase of the serogroup 33 NVT strain compared with saline in mice immunized with PCV13. A statistical difference between the loads of the VT and NVT strains in the same animal was observed in this group, indicating a competitive advantage for the NVT strain in PCV13-immunized mice. In the case of the second mixture, no reduction was observed for the serotype 23F VT strain, which was probably due to the lower levels of anti-23F PS IgG induced by PCV13. Indeed, no binding of IgG antibodies to the serotype 23F strain was observed for sera of mice immunized with PCV13.

Mice nasally immunized with wP or PspA1 + PspA4 + wP showed a reduction of VT and NVT strains after challenge with both mixtures of strains. A statistically significant reduction compared with wP was observed for animals immunized with PspA1 + PspA4 + wP for strains NVT33 (PspA1), VT23F (PspA2), and NVT15/B/C (PspA4), but not for VT4 (PspA3), which may indicate the need for incorporating another PspA clade in the formulation in order to achieve broader protection. These results also show that protection elicited by the protein formulation involves both PspA-specific responses and responses due to wP. A reduction of pneumococcal colonization induced by nasal immunization with wP has already been observed by our group [18,23]. Nasal wash samples from mice inoculated only with wP showed higher IL-6 levels before challenge and an early increase in CXCL-1 one day after challenge with pneumococci [18], which probably contributed to the reduction in the bacterial load in the nasopharynx, possibly by the early recruitment of phagocytic cells. Other groups have also shown a reduction of colonization with pneumococci by nasal adjuvants. Cholera-toxin B-subunit (CTB) used as an adjuvant for a nasal whole-cell pneumococcal vaccine was shown to reduce pneumococcal colonization [24]. The effect of CTB was associated with the activation of a local innate response, involving activation of the caspase-1/11 inflammasome, mucosal T cells, and macrophages [25].

Interestingly, the combination of the PspA nasal formulation with PCV13 led to a reduction of both strains in the two mixtures tested. These results indicate that such combination may be a strategy to overcome serotype replacement observed with the use of PCVs. Animals immunized with the combination elicited both anti-PS and anti-PspA antibodies, which would help reduce the competitive advantage of the NVT strains over VT strains in the co-colonization challenge. Immune response against wP was not measured in this work, but there was a clear effect of the adjuvant by itself in the reduction of the bacterial load. In the case of the mixture of VT4 (PspA3) and NVT33 (PspA1), the higher recovery of the NVT strain compared to the VT strain observed in the PCV13 group was not seen in any other of the immunized groups, wP, PspA1 + PspA4 + wP, and (PspA1 + PspA4 + wP) + PCV13.

Anti-PS antibodies were proposed to protect against carriage acquisition in PCV-vaccinated volunteers through agglutination [26]. Agglutinating antibodies would inhibit the establishment of mucosal colonization due to more efficient mucociliary clearance of larger particles. Using a model of passive transfer of hyperimmune sera raised against encapsulated bacteria, it was proposed that serum IgG antibodies extravasated from plasma to the nasal lumen, protecting mice against pneumococcal nasopharyngeal carriage acquisition by agglutination of bacteria [27]. This protection was shown to be serotype specific. On the other hand, protein antigens were proposed to protect against pneumococcal colonization both through humoral and cellular immune responses. Th17 CD4+ T cells were shown to mediate the clearance of pneumococcal colonization in mice [28,29]. Recent work using an experimental colonization challenge in humans failed to find an association between Th17 or any CD4+ T memory cells and the control of colonization [30,31]. Rather, individuals with a higher concentration of CXCL10 prior to challenge were shown to have higher bacterial loads [30]. Furthermore, B cells were shown to be depleted in the nasal mucosa following establishment of carriage due to recirculation of activated B cells. CD8+ mucosa-associated invariant T (MAIT) cell responses were associated with protection against pneumococcal carriage [31].

In this work, we have only evaluated the induction of the humoral response against PspA, and both serum and mucosal antibodies were detected in mice immunized with the nasal protein vaccine formulation. It is not clear which mechanism would be responsible for the protection elicited by anti-PspA antibodies in our model. Serum anti-PspA IgG antibodies were shown to fix complement on the surface of pneumococci [32], leading to opsonophagocytosis. More recently, agglutination mediated by anti-PspA antibodies was also proposed as a mechanism of protection against colonization [33], similar to anti-PS antibodies. As mentioned before, the adjuvant wP by itself was able to lead to some reduction of bacterial loads and is an important component of the protein vaccine formulation.

There are some caveats related to our co-colonization models. Firstly, children that become colonized with two or more strains may not be exposed and colonized at the same time with these strains, and we have used a model in which mice are inoculated simultaneously with both strains. The dynamics of carriage acquisition of two strains may be sequential rather than at the same time, and a model involving exposure to pneumococci from different serotypes showed that carriage of one strain may inhibit the acquisition of another strain [16]. This adds more complexity to the evaluation of the protective efficacy of vaccines against colonization with different pneumococcal strains. Secondly, in settings with high PCV13 coverage, there is already low circulation of VT strains. Still, colonization with VT strains was shown to persist, and colonization with NVT was shown to increase in a setting of high pneumococcal carriage burden [34]. Competition between VT and NVT strains may thus remain even in PCV13-vaccinated populations. Our model also did not take into account the possibility of horizontal transfer of the antibiotic-resistance cassettes during co-colonization, which might render strains resistant to both erm and spec. Previous work has indeed shown that recombination can occur in mouse models of co-colonization [35]. Still, such recombination events might render a minor proportion of the recovered bacteria resistant to both antibiotics. Finally, pneumococcal infection models in mice have limitations, as humans are the natural host of these bacteria. Furthermore, PCV13 induces protective responses against 23F strains in children, which was not observed in our model.

In conclusion, using models of co-colonization challenge with VT and NVT strains expressing different PspAs, we showed that the combination of PCV13 with nasal immunization with a formulation containing PspA1 and PspA4 adjuvanted with wP controls the carriage of both VT and NVT strains, overcoming the competitive advantage of the NVT strain when only PCV13 was administered. PCVs are very effective against pneumococcal invasive disease caused by VT strains in children. Rather than being replaced, PCVs could be improved through combinations of different immunization strategies to overcome serotype replacement.

## 4. Materials and Methods

### 4.1. Ethics Statement

This study was performed according to the guidelines outlined by the Brazilian National Council for Control of Animal Experimentation (CONCEA). Experimental protocols were approved by the Ethic Committee on Animal Use of the Butantan Institute (CEUAIB) under protocol number 5824140716 (17/08/2016). Animals were housed under controlled temperature and light cycle (12/12 h, light/dark cycle) conditions with daily monitoring. Food and water were given ad libitum.

### 4.2. Pneumococcal Strains

Strains TIGR4 (VT, serotype 4, PspA3–VT4 (PspA3)), HU368/06 (VT, serotype 23F, PspA2–VT23F (PspA2)), 4431/119 (NVT, serogroup 33, PspA1–NVT33 (PspA1)), and 237/53 (NVT, serogroup 15B/C, PspA4–NVT15B/C (PspA4)) (Table 1) were grown on blood agar plates or in Todd-Hewitt containing 0.5% yeast extract (THY). HU368/06 is a clinical isolate from the University of Sao Paulo Hospital (Sao Paulo, Brazil) [14]. 4431/119 and 237/53 are carriage isolates from the Liverpool School of Tropical Medicine (Liverpool, UK). Stocks were frozen in THY with 25% glycerol and were maintained at −80 °C.

### 4.3. Construction of the Strains Resistant to Erm and Spec

The erm-resistance cassette was amplified from plasmid pE693, using primers ErmF BamHI (5′ TAGGGATCCTTCGTGCTGACTTGCACC 3′) and ErmR XhoI (5′ GTAGCTCGAGAGTAACGTGTAACTTTCCAAATTTACAAAAG 3′). The spec-resistance cassette was amplified from plasmid pE729 using primers SpecF BamHI (5′ TAGGGATCCTAACTATAACTAATAACGTAACGTG 3′) and SpecR XhoI (5′ GTAGCTCGAGTATGCAAGGGTTTATTGTTTTC 3′). The cassettes were inserted between regions comprising 250–1250 bp and 1750–2750 bp of TIGR4 *iga*. The gene fragment comprising 250–2750 bp of *iga* was first cloned in pGEM-TEasy (Promega) using primers Iga 250F (5′ GAGGAATAATGGAAAAGTATTTTG 3′) and Iga 2750R (5′ TTTTCACCGATTAAACGAC 3′). The region containing 250–1250 bp, pGEM-TEasy, and 1750–2750 bp was then amplified using primers Iga 1750F XhoI (5′ GTAGCTCGAGGTCCAGAAAAAACTGAAGAAG 3′) and Iga 1250R BamHI (5′ TAGGGATCCTGAAAATCTATTTTTGTCTCTATAG 3′). Resistance cassettes were then inserted between 250–1250 bp and 1750–2750 bp of *iga*. Pneumococcal strains were then transformed with the constructed plasmids using CSP1 and CSP2. Clones were selected in blood agar plates containing erm (1 µg/mL) or spec (300 µg/mL). Insertion of the cassettes in the genomic DNA within *iga* was confirmed by PCR. Serotype/serogroup of the obtained clones was confirmed by PCR serotyping [36]. Expression of PspA by the obtained clones was confirmed by western blot.

### 4.4. Co-Colonization Challenge

Five- to seven-week-old C57BL/6 female SPF mice were obtained from the Medical School of the University of Sao Paulo (FM-USP, Sao Paulo, Brazil). Mice were anesthetized through the intraperitoneal (ip) route with a xylazine/ketamine solution (20 µg/g and 50 µg/g, respectively), and a mixture containing two strains of pneumococci (5 × 10^5^ CFU each) was inoculated intranasally in a volume of 10 μL into both nostrils. Animals did not show any sign of sickness after challenge. For nasal washes, mice were euthanized through the ip route with a lethal dose of a xylazine/ketamine solution (60 µg/g and 300 µg/g, respectively) five days after the challenge. A catheter was inserted into the trachea of the mice, and the upper respiratory tract was rinsed with 200 µL of saline. An additional rinse with 200 µL of saline was performed; volumes were pooled and serially diluted for plating in blood-agar containing gentamycin (4 µg/mL), erm or spec. Pneumococcal loads were determined taking into account CFU recovered on the plates and total volume collected in the nasal washes. The limit of detection was 40 CFU. Samples without recovery of bacteria were plotted as 10 CFU.

### 4.5. Immunization

Protein immunization was performed through the nasal route in saline. Groups of six to eight animals were given three doses of 4 µg of protein (2.0 µg of PspA1 and 2.0 µg of PspA4) containing 1/16 of the human dose of the whole-cell pertussis vaccine wP with a 14-day interval. Recombinant proteins PspA1 and PspA4 comprise the mature α-helical region plus the proline-rich region of PspA (Figure 5a) and were purified by affinity chromatography [14]. Proteins were treated with Triton X-114 to reduce the endotoxin content [37] and with SM-2 beads (Biorad) to remove the Triton X-114 residual content [38]. wP used in this work is composed of whole-cell pertussis inactivated with 0.2% formalin and was produced by Instituto Butantan (São Paulo, Brazil). Our group has previously used wP as an efficient nasal adjuvant for immunization with PspA [18,23]. Vaccines were administered in a volume of 10 μL using a micropipette. Nasal immunization was conducted in mice previously anesthetized through the ip route with a xylazine/ketamine solution (20 µg/g and 50 µg/g, respectively). Three doses containing 1/20 of the human dose of PCV13 were given to mice subcutaneously in a volume of 100 µL. Dose–response analysis of a similar conjugate vaccine has shown the induction of high antibody titers and protection of mice against a lethal challenge with a 6B strain by the inoculation of three doses of 1/25 of the human dose [39]. Serum was collected two weeks after the third immunization for the evaluation of antibody levels. Vaginal washes were collected from days 15 to 19 after the third immunization by pipetting 25 μL of saline twice. Samples from the five days were pooled for each animal. Co-colonization challenge was performed three weeks after the third immunization. The scheme used for immunization, challenge, and collection of sera, vaginal washes, and nasal washes is shown in Figure 5b.

### 4.6. Measurement of Anti-PspA Antibodies by ELISA

ELISA against PspA was carried out as described previously [14] in high-binding plates coated with 1 μg/mL PspA1 or PspA4. For the detection of serum antibodies, goat anti-mouse IgG conjugated with horseradish peroxidase (HRP, Sigma) was used as a secondary antibody. The titer was defined as the reciprocal of the highest dilution with an Abs 492 nm ≥ 0.1. For the detection of IgG and IgA in vaginal washes, goat anti-mouse IgG conjugated with HRP and goat anti-mouse IgA conjugated with HRP (Sigma) were used, respectively. Abs 492 nm for samples diluted 1:2 (IgG) or pure samples (IgA) is shown.

### 4.7. Measurement of Anti-PS Antibodies by ELISA

Medium-binding plates were coated with 10 µg/mL PS4, PS15B, PS23F or PS33 (American Type Culture Collection). Serum samples were incubated with PS22F (5 µg/mL) prior to incubation in the plate coated with PS. Anti-PS IgG was detected with anti-mouse IgG conjugated with HRP. The titer was defined as the reciprocal of the highest dilution with an Abs 492 nm ≥ 0.1.

### 4.8. Binding of Antibodies to Intact Bacteria

Antibody-binding assay was performed as previously described [32]. Briefly, pneumococci were plated on blood agar for overnight growth, then cultured in THY to OD 600 nm 0.4–0.5 (~10^8^ CFU/mL) and harvested by centrifugation. Bacteria were washed, suspended in PBS, and incubated with 1% of individual sera for 30 min at 37 °C. Samples were washed once with PBS before incubation with fluorescein isothiocyanate (FITC)-conjugated anti-mouse IgG (Sigma) for 30 min on ice. Samples were fixed with 2% formaldehyde after two washing steps and stored at 4 °C. Flow cytometry analysis was conducted using FACSCanto (BD Biosciences), and 10,000 gated events were recorded. Mean fluorescence intensity (MFI) was determined for each sample.

### 4.9. Statistical Analysis

Differences in the colonization with erm- and spec-resistant clones in the same sample in naïve mice were evaluated by Student’s paired *t*-test. Differences in colonization between immunization groups were evaluated by the Mann–Whitney Test. Differences in antibody levels and MFI between groups were analyzed using one-way ANOVA with Tukey’s multicomparison test. GraphPad Prism 8.1.2 (San Diego, CA, USA) was used for statistical analysis.

## Figures and Tables

**Figure 1 pathogens-09-00278-f001:**
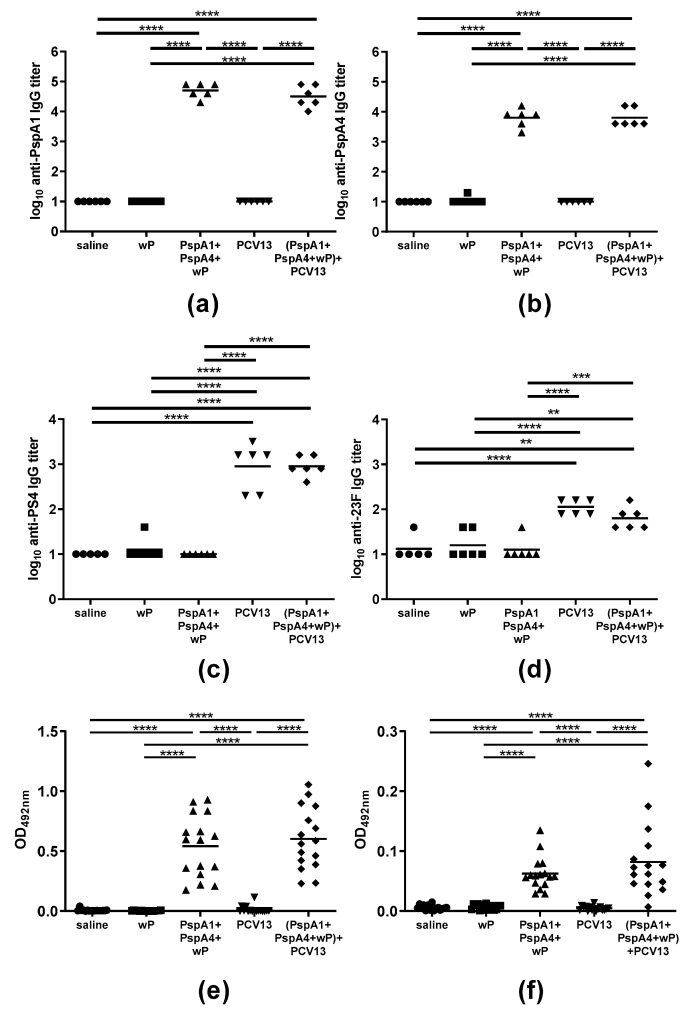
Induction of antibodies. Mice received three doses of the indicated vaccine formulations and serum IgG antibodies against (**a**) PspA1, (**b**) PspA4, (**c**) PS4, and (**d**) PS23F and vaginal **(e)** IgG and **(f)** IgA antibodies against PspA1 were measured by ELISA. * indicates statistical difference (One-way ANOVA, Tukey’s multicomparison Test- ** *p* ≤ 0.01; *** *p* ≤ 0.001; **** *p* ≤ 0.0001). Results are representative of two independent experiments.

**Figure 2 pathogens-09-00278-f002:**
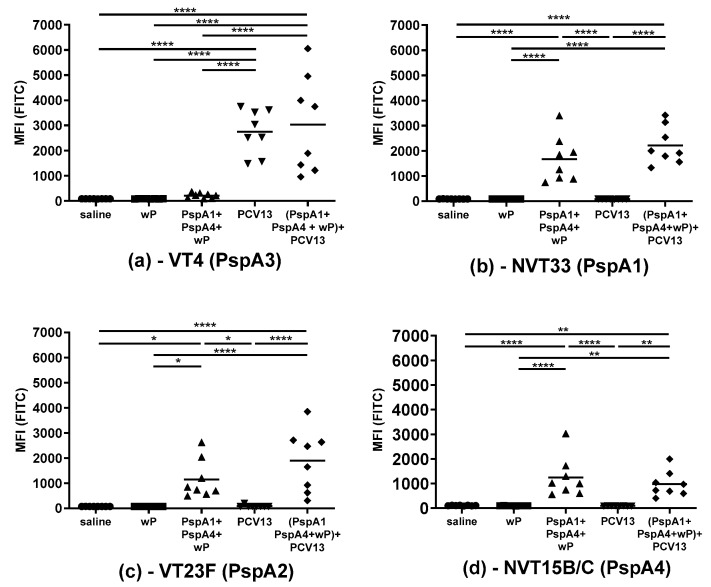
Binding of serum IgG to the surface of pneumococci. Individual sera from mice immunized with three doses of the indicated formulations were tested for the ability to bind to pneumococcal strains (**a**) VT4 (PspA3), (**b**) NVT33 (PspA1), (**c**) VT23F (PspA2), and (**d**) NVT15B/C (PspA4). Results are shown as mean fluorescence intensity (MFI) * indicates statistical difference (One-way ANOVA, Tukey’s multicomparison Test- * *p* ≤ 0.05; ***p* ≤ 0.01; **** *p* ≤ 0.0001).

**Figure 3 pathogens-09-00278-f003:**
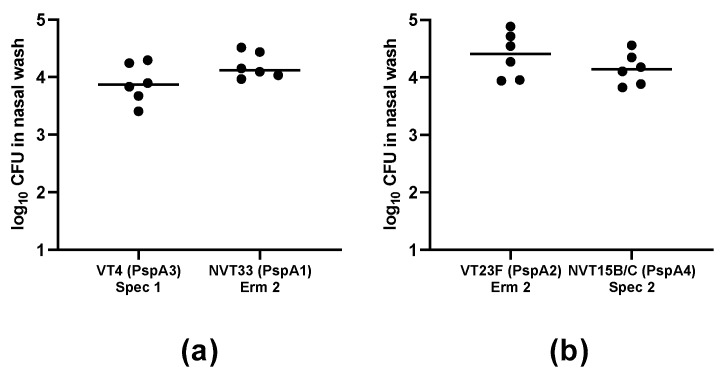
Co-colonization models. Naïve mice were inoculated with a mixture of strains (**a**) VT4 (PspA3) Spec 1 + NVT33 (PspA1) Erm 2 and (**b**) VT23F (PspA2) Erm 2 + NVT15B/C (PspA4) Spec 2, and recovery of erm- and spec-resistant pneumococci was evaluated in nasal washes collected after five days. Differences in the recovery of erm- and spec-resistant bacteria in the same sample were not significant based on Student’s paired *t*-test.

**Figure 4 pathogens-09-00278-f004:**
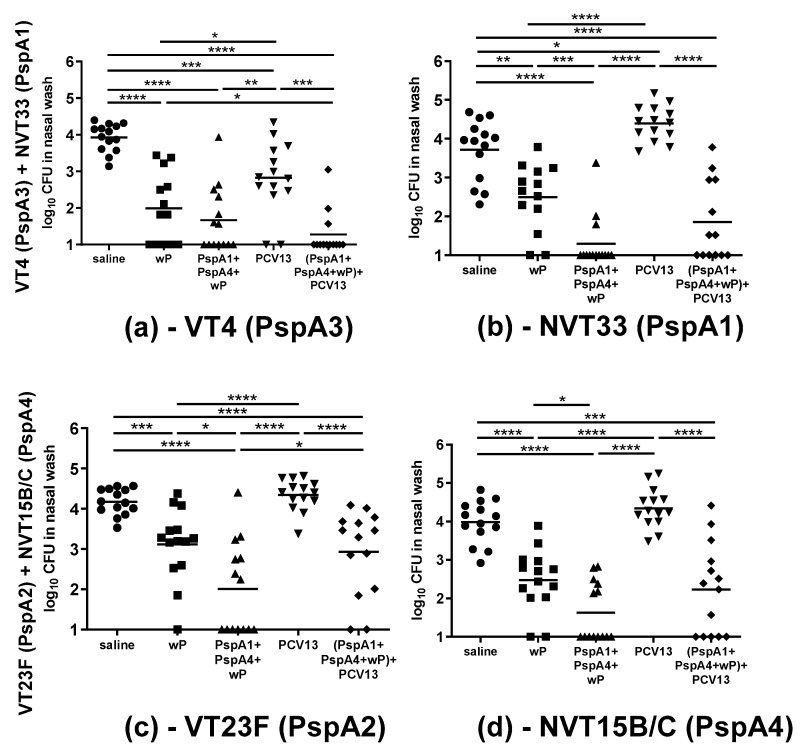
Co-colonization challenge. Mice were immunized with three doses of the indicated formulations and challenged with the mixture of strains (**a**,**b**) VT4 (PspA3) + NVT33 (PspA1) or (**c**,**d**) VT23F (PspA2) + NVT15B/C (PspA4). The recovery of strains (**a**) VT4 (PspA3), (**b**) NVT33 (PspA1), (**c**) VT23F (PspA2), and (**d**) NVT15B/C (PspA4) in nasal washes performed five days after challenge is shown. * indicates a statistical difference (Mann–Whitney Test; * *p* ≤ 0.05; ** *p* ≤ 0.01; *** *p* ≤ 0.001; **** *p* ≤ 0.0001). Results are from two independent experiments.

**Figure 5 pathogens-09-00278-f005:**
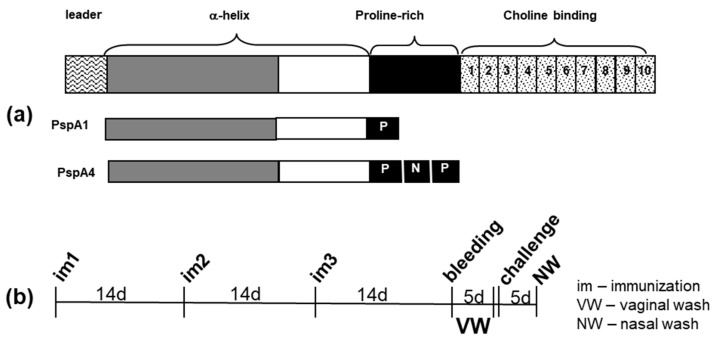
Recombinant proteins and immunization scheme. (**a**) The regions of PspA present in the recombinant proteins PspA1 and PspA4 used for immunization and (**b**) the scheme used for immunization, collection of blood, vaginal washes, co-colonization challenge, and collection of nasal washes is shown.

**Table 1 pathogens-09-00278-t001:** Pneumococcal isolates.

Isolate	Identification	Serotype/Serogroup	VT/NVT	PspA Family	PspA Clade
TIGR4	VT4 (PspA3)	4	VT	2	3
368/06	VT23F (PspA2)	23F	VT	1	2
4431/119	NVT33 (PspA1)	33	NVT	1	1
237/53	NVT15B/C (PspA4)	15B/C	NVT	2	4

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
