# Peer review of "Efficacy of a Protein Vaccine and a Conjugate Vaccine Against Co-Colonization with Vaccine-Type and Non-Vaccine Type Pneumococci in Mice"

_pathogens, 2020, doi:10.3390/pathogens9040278_

Round 1
Reviewer 1 Report
This manuscript assesses a combination of PCV and protein vaccine approaches to protect mice against colonization with different strains of pneumococci. Overall, the paper is well written and data clearly presented.
Specific comments
- In my view, the authors somewhat over interpret their data. For example, the data really don’t suggest that the anti-PS responses and anti-protein responses interact with one another, rather they are independent events. For example, the best protection against all 4 strains is provided by PspA1+PspA4+wP. PCV13 provides no additional additive or synergistic protection. Thus, the final conclusion (lines 261-266) that a “combination of different immunization strategies may thus be an alternative to overcome serotype replacement caused by PCVs” could easily be replaced by an alternative interpretation that a protein vaccine can completely replace PCVs and potentially avoid serotype replacement. At least in mice…
- I also disagree with the authors assertion that the data indicates a “competitive advantage for the NVT strain induced by PCV13 vaccination” (lines 154-155) and repeated in lines 261-266. To me the impact of PCV13 against the VT is completely independent of the lack of effect against the NVT (and results likely would have been the same if mice were challenge by the individual strains rather than co-colonized). A competitive advantage would only be demonstrated if the niche created by reduction of the VT was filled by the NVT and this was certainly not demonstrated.
- I think that the fact that wP alone is a better vaccine than PCV13 against both NVT and VT warrants more discussion, particularly the relevance of this model to humans, where presumably this is not the case. Some discussion of the limitations of the mouse model could be added to the second last paragraph of the discussion. Also in the context of the poor anti-23F response elicited by PCV13.
Minor comments
- Lines 47-48 – don’t use alternative twice – the 2nd one can be removed.
- The authors should be careful, as much as possible, to distinguish between results and discussion. For example, paragraph 2 of the discussion is primarily reiteration of the results and not discussion of the results.
Author Response
1. In my view, the authors somewhat over interpret their data. For example, the data really don’t suggest that the anti-PS responses and anti-protein responses interact with one another, rather they are independent events. For example, the best protection against all 4 strains is provided by PspA1+PspA4+wP. PCV13 provides no additional additive or synergistic protection. Thus, the final conclusion (lines 261-266) that a “combination of different immunization strategies may thus be an alternative to overcome serotype replacement caused by PCVs” could easily be replaced by an alternative interpretation that a protein vaccine can completely replace PCVs and potentially avoid serotype replacement. At least in mice…
We agree with the reviewer that we did not show a synergistic effect of anti-protein and anti-PS responses. Our suggestion of using combination vaccines is based on the fact that PCVs are very efficient against invasive pneumococcal disease caused by vaccine-type strains in children and it is mostly agreed that it will not be possible to replace PCVs. Rather, improvement of current PCVs should be pursued. We have added a comment on that in lines 301-304.
2. I also disagree with the authors assertion that the data indicates a “competitive advantage for the NVT strain induced by PCV13 vaccination” (lines 154-155) and repeated in lines 261-266. To me the impact of PCV13 against the VT is completely independent of the lack of effect against the NVT (and results likely would have been the same if mice were challenge by the individual strains rather than co-colonized). A competitive advantage would only be demonstrated if the niche created by reduction of the VT was filled by the NVT and this was certainly not demonstrated.
We agree with the reviewer that to demonstrate a competitive advantage, there should be an increase in the NVT strain in PCV13 immunized mice. We have altered the statistical analysis to non-parametric Mann-Whitney test as suggested by reviewer #3 and we observed an increase in the NVT strain in PCV13 immunized mice compared to saline with borderline significance. We have indicated this in the text (lines 171-172).
3. I think that the fact that wP alone is a better vaccine than PCV13 against both NVT and VT warrants more discussion, particularly the relevance of this model to humans, where presumably this is not the case. Some discussion of the limitations of the mouse model could be added to the second last paragraph of the discussion. Also in the context of the poor anti-23F response elicited by PCV13.
We have added comments on the limitations of mouse models of pneumococcal infection to the discussion (lines 293-296).
Minor comments
1. Lines 47-48 – don’t use alternative twice – the 2nd one can be removed.
The sentence has been corrected.
2. The authors should be careful, as much as possible, to distinguish between results and discussion. For example, paragraph 2 of the discussion is primarily reiteration of the results and not discussion of the results.
We agree with the reviewer that we should distinguish results and discussion as much as possible. Still,we tried in some paragraphs to describe our results before adding related data from the literature for clarity.
Reviewer 2 Report
Very good analysis comparing the vaccine type strains with the non vaccine type strains and the effects of the pneumococcal conjugate vaccines and the combination pneumococcal protein/pneumococcal conjugate vaccines.
A comment made was the observation that the co-colonization models were set up using simultaneous exposure of the different strains of pneumococci. It is believed that invivo exposure will most likely involve sequential exposure of different strains of pneumococci. Were there any experiments set up (or planned) to address this? The authors made a good point here and perhaps should pursue this.
Author Response
We thank the reviewer for the comments. It would indeed be important to test how the vaccines, both PCV13 and the nasal protein formulation, affect co-colonization when the challenge is performed sequentially with a vaccine type strain and then with a non-vaccine type strain (or vice-versa). Though we did not plan such experiments yet, we agree that it would be worth pursuing this. The paper on horizontal transfer indicated by reviewer #2 (Marks et al 2012) actually shows a model of sequential colonization and could be adapted for analyzing vaccine efficacy.
Reviewer 3 Report
In this paper, Colichio and colleagues present an interesting study exploring the dynamics of serotype replacement under different vaccine pressures during a mouse model of co-colonisation. As to be expected, they identified that PCV13 vaccination was associated with a decline in vaccine type density in the nasopharynx whereas non-vaccine type pneumococci were unaffected. However, combined immunization of PCV13 with a protein-based vaccine targeting major virulence factor PspA resulted in both strains being affected, suggesting this may be a promising strategy to reduce serotype replacement. Overall, this study makes a valuable contribution to the field, but I do have some comments to improve the manuscript:
- The co-colonization model described in this study involves infecting mice with two different pneumococcal strains (e.g. strain x and y) that have different markers (erm or spec) so that plating nasal wash samples on media with either erm or spec will allow you to select for and quantify the strain of interest. However, it is well established that pneumococci are naturally competent and readily participate in horizontal gene transfer. Additionally, it has been shown that transformation of pneumococci occurs in the nasopharynx, the site examined in this study. Therefore, it is more than plausible that horizontal transfer of the erm and spec markers can occur between these strains and as a result, strains can become resistant to both antibiotics and would grow on both types of selective media. As such, when the authors plate out their nasal wash on spec for example, expecting to get growth of strain x only, it is likely that a proportion of colonies on that plate would be recombinant strain y that has taken up the spec marker from strain x. Of course, this is a somewhat rare occurrence, but previous studies have shown this is possible by setting up a very similar study design to that described in this manuscript; a co-colonization mouse model and looking at exchange of antibiotic resistance markers in the nasopharynx (which did in fact occur at a transformation efficiency of 10-2) (see Marks et al 2012 mBio:3(5)e00200-12). I think this is an important limitation of the study that should either be tested (i.e. plating nasal wash samples on media with spec, erm and spec+erm to determine how many are actually recombinants and not the original strains and whether vaccination affects this process) or by acknowledging this as a limitation of their study in the discussion.
- Statistics – could the authors please provide some rationale for the statistical tests chosen? My concerns are around the pneumococcal count data. In figure 1, pneumococcal viable counts were log transformed (which is the correct thing to do). However, data were analyzed by student’s t test. As pneumococcal count data are not normally distributed, I do not understand how this is an appropriate test. Instead, as the data are not normally distributed, they should be presented as the median (error bars = interquartile range) and analyzed by Mann-Whitney U test as can be seen in several other papers analyzing pneumococcal count data from mouse tissues. Likewise, I see a similar issue with figure 4.
- Other comments:
- Line 37: PCV is an abbreviation for Pneumococcal Conjugate Vaccine, not polysaccharide
- I would suggest swapping results sections 2.1 and 2.2. Currently the results section reads as co-colonization model, response to vaccine (in the absence of challenge) and then back to co-colonization model
Author Response
- We agree with the reviewer that there could recombinants that became resistant to both antibiotics during co-colonization and we did not take this into account while performing the experiments. Thus, we do not have the data for recovery of bacteria resistant to both erm and spec. We did plate the samples in gentamicin only and we did not observe any major discrepancies in the data. Of course, this only tell us that the majority of the colonies would be resistant either to erm or spec. Furthermore, the size of the colonies on agar plates varies with the strain and we could usually discriminate VT and NVT strains used in the co-colonization models just by looking at the blood agar plates containing gentamicin. Still, we agree that this is a shortcoming of our model and we have acknowledged this in the text (lines 289-294).
- Statistics. We agree with the reviewer. Though data on colonization of naïve mice is normally distributed, CFU data in the immunized groups do not have normal distribution, especially for groups with samples with no recovery of bacteria and plotted as 10 CFU. Thus, we have maintained Student´s paired t-test for naïve mice in Figure 3 (formerly Figure 1) and altered analysis of data in Figure 4 to Mann-Whitney test (non-parametric, unpaired) and in Supplementary Figure 1 to Wilcoxon matched-paired rank test (non-parametric, paired).
- Line 37: PCV is an abbreviation for Pneumococcal Conjugate Vaccine, not polysaccharide. We have corrected the sentence in line 37.
-
I would suggest swapping results sections 2.1 and 2.2. Currently the results section reads as co-colonization model, response to vaccine (in the absence of challenge) and then back to co-colonization model. We have swapped sections 2.1 and 2.2
Round 2
Reviewer 3 Report
I am satisfied with how the authors have addressed my comments.